# Physiological and Transcriptome Responses of *Pinus massoniana* Seedlings Inoculated by Various Ecotypes of the Ectomycorrhizal Fungus *Cenococcum geophilum* during the Early Stage of Drought Stress

**DOI:** 10.3390/jof10010071

**Published:** 2024-01-15

**Authors:** Xiaohui Zhang, Jinyan Zhang, Juan He, Mingtao Li, Norihisa Matsushita, Qifang Geng, Chunlan Lian, Shijie Zhang

**Affiliations:** 1International Joint Laboratory of Forest Symbiology, College of Forestry, Fujian Agriculture and Forestry University, Fuzhou 350002, China; m15059255681@163.com (X.Z.); zjy3297@163.com (J.Z.); hejuanhena@163.com (J.H.); 17720799223@163.com (M.L.); 2Graduate School of Agricultural and Life Sciences, The University of Tokyo, 1-1-1 Yayoi, Bunkyo-ku, Tokyo 113-8657, Japan; nmatsushita@g.ecc.u-tokyo.ac.jp; 3College of Forestry, Shandong Agricultural University, No. 61 Daizong Street, Taian 271018, China; gengqf@sdau.edu.cn; 4Asian Research Center for Bioresource and Environmental Sciences, Graduate School of Agricultural and Life Sciences, The University of Tokyo, 1-1-1 Midori-cho, Tokyo 188-0002, Japan; 5Jiangsu Key Laboratory for the Research and Utilization of Plant Resources, Institute of Botany, Jiangsu Province and Chinese Academy of Sciences (Nanjing Botanical Garden Mem. Sun Yat-Sen), No. 1 Qianhuhoucun, Zhongshanmen, Xuanwu District, Nanjing 210014, China

**Keywords:** *Pinus massoniana*, *Cenococcum geophilum*, drought stress, antioxidant activity, mycorrhiza

## Abstract

The impact of drought stress on plant growth in arid regions is a critical concern, necessitating the exploration of strategies to enhance plant drought resistance, particularly during the early stages of drought stress. This study focuses on the ectomycorrhizal fungus *Cenococcum geophilum*, renowned for its extensive genetic diversity and broad host compatibility, making it a crucial ally for host plants facing external stresses. We utilized *Pinus massoniana* seedlings inoculated with different ecotypic strains of *C. geophilum* under drought stress. The results showed that the inoculation of most strains of *C. geophilum* enhanced the drought resistance of *P. massoniana* seedlings under the early stages of drought stress, by influencing the water content, photosynthesis, accumulation of osmotic adjustment substances, and antioxidant enzyme activities in both shoots and roots of seedlings. Transcriptome analysis showed that mycorrhizal seedlings mainly regulated energy metabolism and reduction–oxidation reaction to resist early drought stress. Notably, the level of drought resistance observed in mycorrhizal seedlings was irrespective of the level of drought tolerance of *C. geophilum* strains. This study contributes essential data for understanding the drought response mechanisms of mycorrhizal *P. massoniana* seedlings inoculated by distinct *C. geophilum* ecotypes and guidance on selecting candidate species of ectomycorrhizal fungi for mycorrhizal afforestation in drought areas.

## 1. Introduction

With the changing climate, the global reduction in precipitation has resulted in an increased occurrence of droughts [1]. Drought stress can significantly influence plant root growth by impeding elongation and branching, ultimately diminishing a plant’s capacity to absorb water and nutrients in soils. Furthermore, it can lead to reduced photosynthesis and nutrient accumulation, cellular damage, and, in the end, plant wilting and mortality [2,3]. Notably, it has been documented that from 2010 to 2017, drought events were responsible for the demise of approximately 129 million trees in California [4]. Likewise, severe drought has resulted in widespread mortality of *Pinus tabulaeformis* in central and eastern China [5]. The global impact of drought on plant growth and crop yields is undeniable, and climate-induced and soil-related droughts are projected to intensify further due to ongoing global climate change [6]. When plants experience drought stress, their phenotypic properties and physiological traits can undergo responses, and the degree to which each property responds in the early stage of drought is closely linked to the drought resistance of the plants [7]. Consequently, it is imperative to explore strategies aimed at enhancing the drought resistance of plants under the early stage of drought stress. *Pinus massoniana* stands as one of the most vital timber resources in southern China, distinguished by its rapid growth, high yield, and multifaceted utility, all while contributing significantly to ecological value [8]. Nonetheless, during the early stages of afforestation, the survival rate of *P. massoniana* seedlings is notably hindered by factors such as inadequate shade and soil dryness [9]. Therefore, it becomes imperative to enhance the drought resistance of *P. massoniana* seedlings.

Ectomycorrhiza (ECM) represents a mutualistic symbiosis formed between ectomycorrhizal fungi (ECMF) and non-lignified absorbing roots [10]. ECMF can establish symbiotic relationships with plant families such as Pinaceae, Fagaceae, Salicaceae, and Betulaceae [11,12,13,14]. Under conditions of drought stress, ECM demonstrates the capacity to expand the root absorption zone, thereby enhancing water and nutrient uptake through external hyphae. This expansion results in improved root water retention and notable enhancements in root traits, including increased root biomass, total length, average diameter, and the number of root tips in the host plant. These factors collectively play a crucial role in augmenting drought resistance [15]. Furthermore, Zou et al. [16] have observed that under water-deficient conditions, ECM hyphae can extend into soil gaps, providing additional water resources to mitigate drought effects. Additionally, ECM symbiosis contributes to increased water and nutrient absorption, facilitates photosynthesis, regulates the synthesis of antioxidant enzymes, and facilitates the accumulation of osmotic substances within plants [17]. Notably, Ahmed et al. [18] have reported that ECMF promotes the formation of osmotic adjustment substances, such as plant-soluble proteins, thereby maintaining cellular water balance and enhancing plant drought tolerance. In a study by Zhao et al. [19], ECMF inoculation was found to elevate antioxidant enzyme activity in *P. sylvestris*. These findings collectively underscore the significant role of ECM symbiosis in enhancing plant resilience to drought, a phenomenon that holds paramount importance in the face of increasing environmental challenges. Therefore, the investigation of the influence of the different ECMF species on *P. massoniana* drought resistance, and the underlying molecular mechanisms governing, is indispensable to the early stages of afforestation of *P. massoniana*.

*Cenococcum geophilum* Fr., a commonly encountered ectomycorrhizal fungus in natural ecosystems, typically exists in soil in mycorrhizal and sclerotial forms, without the formation of spores, and is characterized by a profusion of external hyphae [20]. This fungal species exhibits a broad spectrum of host compatibility and displays robust adaptability and stress resistance, making it a valuable ally for plant growth across diverse and challenging environments [21]. Notably, *C. geophilum* demonstrates exceptional drought tolerance and is often employed as a prominent candidate in water-scarce forest ecosystems. Research by Coleman et al. [22] has illuminated its role in preserving cellular integrity under conditions of water deficit while mitigating the surge in reactive oxygen species (ROS) typically observed during drought stress. Furthermore, recent findings by Li et al. [23] underscore the ability of different ecotypes of *C. geophilum* to induce the expression of drought-related genes when subjected to drought conditions, thereby enhancing drought resistance. It is important to note that different ecotypes of *C. geophilum* exhibit distinct responses to drought, presenting an intriguing avenue for further exploration in this context.

In this study, we undertook the inoculation of *P. massoniana* seedlings with eight ecotypes of *C. geophilum*. By meticulously simulating the early stage of drought stress conditions for 7 days, we aimed to investigate the diverse impacts of these ecotypes of *C. geophilum* on the growth and physiological responses of *P. massoniana* seedlings when subjected to the early stage of drought stress. Subsequently, we selected mycorrhizal seedlings that performed better (drought-tolerant) and worse (drought-sensitive) under the early stage of drought stress for in-depth RNA sequencing. The comparative assessment of differentially expressed genes in drought-tolerant and drought-sensitive mycorrhizal seedlings under the early stage of drought stress provided valuable insights into the physiological and molecular mechanisms underlying the capacity of various ecotypes of *C. geophilum* to enhance the drought tolerance of *P. massoniana*. This research endeavor aims to furnish a robust theoretical foundation for future afforestation efforts involving mycorrhizal seedlings, ultimately contributing to more resilient ecosystems in the face of drought challenges.

## 2. Materials and Methods

### 2.1. The Mycelial Growth of Different Ecotypes of C. geophilum under Drought Treatment

Eight ecotypic strains of *C. geophilum* (Jacg16, Jacg21, Jacg37, Jacg81, Jacg121, Jacg189, Jacg243, Chcg57) were used in this study. The strains in Japan were isolated by the Laboratory of Forest Symbiosis at the Graduate School of Agricultural and Life Sciences, University of Tokyo, while the strain in China was provided by the International Joint Laboratory of Forest Symbiosis at Fujian Agriculture and Forestry University. The source information of each strain is shown in Appendix A. The internal transcribed spacer (ITS) regions of eight ecotypes of *C. geophilum* were amplified using the fungal-specific primer pair ITS1f and ITS4 [24], and sequenced, confirming their belonging to the same species (Appendix A). The mycelia of *C. geophilum* strains were pre-cultured in a modified Melin-Norkrans (MMN) agar medium [25] at 25 °C in the dark for 30 days. Subsequently, an agar plug of *C. geophilum* with a 7 mm diameter was transferred to the center of a 90 mm petri dish, which was lined with a cellophane membrane at the bottom. Each dish contained 10 mL of MMN liquid medium with 0% or 10% polyethylene glycol (PEG-6000, simulated drought) and then cultured in the dark at 25 °C. Three replicates of each strain were conducted for each treatment. After culturing for 30 days, the mycelial growth area of each strain was measured by X-Plan 380dⅢ, Ushikata (Kantum Ushikata Co., LTD., Yokohama, Japan).

### 2.2. Preparation of Mycorrhizal Seedlings of P. massoniana

*Pinus massoniana* seeds were obtained from the Wuyi National Forest Farm in Fujian Province. These seeds were sterilized in 1% sodium hypochlorite (NaCIO, *v*/*v*) for 10 min, followed by rinsing with sterile deionized water five times. The sterilized seeds were then sown in sterilized vermiculite and cultured in a controlled environment for 6 weeks (25 °C, 16 h light cycle). Simultaneously, the eight *C. geophilum* strains were pre-cultured in MMN agar medium at 25 °C in the dark for 45 days.

The substrate used for planting the mycorrhizal seedlings of *P. massoniana* inoculated by *C. geophilum* consisted of a mixture of forest soil and red jade soil (1:2, *v*/*v*). The forest soil was collected from the Soil and Water Moisturizing Garden of Fujian Agriculture and Forestry University in Fujian Province, China. The mixture was then sterilized by autoclaving at 121 °C for 3 h. Six-week-old seedlings of *P. massoniana* with consistent growth were transplanted into a rectangle rhizobox (23.5 × 8.5 × 1.6 cm), and then *C. geophilum* plugs were inoculated onto the lateral roots, while the roots of non-inoculated seedlings were covered with the agar plugs with the same size containing MMN medium. These seedlings were placed in a growth chamber with a light cycle of 16 h at 25 °C. Throughout the cultivation period, the seedlings were weekly applied by 0.1% Hoagland nutrient solution. After 45 days of cultivation, the seedlings of *P. massoniana* with mycorrhizal root rate exceeding 90% were selected for further drought stress experiments.

### 2.3. Experimental Design for Drought Stress of P. massoniana Seedlings

Each of the aforementioned seedlings was transplanted into a 50 mL centrifuge tube filled with 40.0 g of a soil mixture (forest soil:red jade soil = 1:2, *v*/*v*) and then cultivated in a greenhouse under controlled conditions with an average relative humidity of 70% and the illumination duration was set at 16 h at 25 °C. During the cultivation period, the moisture level in the soil was consistently maintained at a range of 85-90% of its field capacity. After 2 months of growth, the seedlings were treated with different drought stress. During the treatment period, each tube with seedlings was weighed and watered daily at 9:00 am to keep the water content in the soil at the required level. For the drought treatment, the field capacity in the soil was reduced to 30-35%, while for the non-drought treatments, the field capacity in the soil was maintained at 85-90%. In each treatment, each seedling inoculated by different ecotypic strains of *C. geophilum* had 15 replicates. Following 7 days of different drought treatments, the seedlings were collected for further analysis.

### 2.4. Determination of Photosynthetic Index

After the 7 days of drought treatments, the gas exchange parameters of *P. massoniana* needles were measured using a photosynthesis instrument with a red-blue light source (LI-6400XT, Lincoln, NE, USA). The parameters measured included the net photosynthetic rate (Pn), transpiration rate (Tr), intercellular CO_2_ concentration (Ci), and stomatal conductance (Gs).

### 2.5. Determination of Morphological and Physiological Indicators

The collected seedlings were cleaned with double distilled water, and the excess water was removed with absorbent paper. Subsequently, three seedlings for each treatment were randomly selected to measure the fresh weight of shoots and roots. The freshly weighed seedlings were then dried at 80 °C until constant weight, and their dry weight was recorded using a digital scale. The water content in both the shoot and root of the seedlings was determined by the gravimetric method. The remaining fresh seedlings were immediately treated with liquid nitrogen and stored in a refrigerator at −80 °C for the determination of osmotic adjustment substances, malondialdehyde (MDA) and antioxidant enzymes. To determinate each index, 0.2 g of fresh shoot or root tissue from each sample was quickly ground into a powder after freezing with liquid nitrogen. The determination of antioxidant enzymes and MDA followed the methods described by Zhang [26]: Catalase (CAT) activity was assessed using the ultraviolet absorption method, peroxidase (POD) activity was determined by employing guaiacol and hydrogen peroxide as substrates in the reaction and measuring the absorbance at 470 nm, superoxide dismutase (SOD) activity was measured using the nitroblue tetrazolium (NBT) reduction method, and the content of MDA in the seedlings was quantified utilizing the thiobarbituric acid (TBA) method. In addition, the proline (Pro) content was determined at 520 nm, following the procedure outlined by Bates et al. [27], while the soluble protein (SP) content was assessed using the Coomassie brilliant blue method as described by Bradford [28].

### 2.6. RNA-Seq and qRT-PCR Analysis

After evaluating the drought tolerance of different ecotypic strains of *C. geophilum* and mycorrhizal seedlings of *P. massoniana* inoculated with these strains of *C. geophilum*, the Jacg121-inoculated (drought-tolerant) and Chcg57-inoculated (drought-sensitive) seedlings were selected for transcriptome analysis. The preparation method and drought stress treatment for these mycorrhizal seedlings remained consistent with the procedures previously described. After 7 days of different drought stress treatment, the shoots and roots of each seedling were frozen in liquid nitrogen and stored at −80 °C. The frozen shoots and roots were sent to Biomarker Technologies (Beijing, China) for total mRNA extraction, cDNA library construction, and sequencing. Briefly, the frozen samples were ground into powder by mortar and pestle and immediately transferred to the lysis extraction buffer. Total mRNA from both the shoots and roots of seedlings was extracted using the RNAsimple total RNA extraction kit provided by Tiangen Biotech (Beijing, China). The cDNA library was constructed, and the sequencing was performed by an Illumina NovaSeq 6000 platform in accordance with standard protocols. The clean reads of sequencing were assembled using Trinity software v2.0.2 to obtain a UniGene library [29]. Differentially expressed genes’ analysis was performed by DESeq2 software 1.42.0 [30]. The novel genes identified in the enrichment analysis were ruled out in further analysis. Additional annotation and functional analysis of identified DEGs were conducted using the Kyoto Encyclopedia of Genes and Genomes (KEGG) and Gene Ontology (GO) databases [31,32], as well as PlantGSEA analysis [33].

To confirm the gene expression levels obtained from the RNA-seq, a real-time quantification polymerase chain reaction (RT-qPCR) analysis was performed. An aliquot of the RNA sample prepared for RNA sequencing was also used for RT-qPCR analysis. The total RNA reverse transcription and RT-qPCR reactions were performed using a Novozyme HiScript II Q RT SuperMix for qPCR (+gDNA wiper) Kit and a ChamQ Uniweisal SYBR qPCR Master Mix Kit (Vazyme Biotech, Nanjing, China). Gene-specific primers were designed based on the UniGene sequence by the tool Premier 5.0 (http://www.premierbiosoft.com/; accessed on 2 June 2022). The Aquaporin protein gene (AQP) was amplified as an internal reference gene to detect the effectiveness of template preparation. The expression of each gene was confirmed in at least three rounds of independent RT-qPCR reactions.

### 2.7. Statistical Analysis

A principal component analysis (PCA) for comprehensively assessing the drought tolerance of plants was employed to evaluate the drought tolerance of mycorrhizal seedlings of *P. massoniana* inoculated by different ecotypic strains of *C. geophilum*, as reported by Zou et al. [34]. This evaluation was based on parameters including water content, photosynthetic indexes, and various physiological indexes.

The formula for calculating the membership function of the comprehensive index of mycorrhizal seedlings inoculated by different ecotypic strains of *C. geophilum* is as below:μXj=(CIj−CImin)(CImax−CImin)

Index weight:Wj=Pj∑j=1nPj

Comprehensive evaluation of drought tolerance level:$$D=\sum_{i=1}^{n}\left(\mu (X_{j})\times W_{j}\right)$$
where *CI_j_* is the *j*th comprehensive index, *CI_min_* is the minimum value of the *j*th comprehensive index, *CI_max_* is the maximum value of the *j*th comprehensive index, and *P_j_* is the contribution index of *CI_j_*.

SPSS (Statistical Product and Service Solutions 21.0) was used for principal component analysis, Pearson correlation analysis, Mann-Whitney U test, and Student’s *t*-test. Pictures were drawn using Origin 2023b software.

## 3. Results

### 3.1. Assessing the Drought Tolerance of Various C. geophilum Ecotypic Strains and Their Impact on the Growth of P. massoniana Seedlings through Inoculation

Under non-drought conditions, eight *C. geophilum* ecotypic strains exhibited variations in their mycelial growth (Figure 1). Jacg37 showed the largest mycelial growth, followed by Jacg189, Chcg57, and Jacg16. Conversely, Jacg21 exhibited the smallest mycelial growth. Comparatively, when cultured to the drought treatment with 10% DEG-6000, the mycelial growth of Jacg16, Jacg21, Jacg37, Jacg81, Jacg121, and Jacg189 was not significantly affected. However, the mycelial growth of Jacg243 and Chcg57 was significantly enhanced and inhibited, respectively.

The growth of *P. massoniana* seedlings non-inoculated and inoculated with eight *C. geophilum* strains is shown in Figure 2. In comparison to non-inoculated seedlings, those inoculated with each *C. geophilum* ecotypic strain exhibited an increase in fresh and dry weight for both shoots and roots. Notably, compared with non-inoculated seedlings, the shoots and roots fresh weights of seedlings inoculated with Jacg121, Jacg37, and Chcg57 were remarkably increased (shoots, 74.00%, 38.00%, and 30.67%; roots, 281.58%, 119.21%, and 99.21%), the shoots fresh weights of seedlings inoculated with Jacg189 and Jacg21 showed a significant enhancement (34.67% and 33.33%), and the roots fresh weights of seedlings inoculated with Jacg81 and Jacg243 remarkably increased by 87.10% and 86.58%, respectively. For dry weights, the shoots and roots of seedlings inoculated with Jacg121 and Chcg57 showed a remarkable increase (shoots, 33.45% and 19.89%; roots, 148.60% and 58.16%), the root dry weights of seedlings inoculated with Jacg37 remarkably increased by 46.93%, when compared to the non-inoculated seedlings.

### 3.2. Effects of Inoculation with Different Ecotypic Strains of C. geophilum on Water Content and Photosynthetic Parameters of P. massoniana Seedlings during the Early Stage of Drought Stress

Compared to non-inoculated seedlings, the water content in shoots of seedlings inoculated with all different *C. geophilum* stains increased significantly by 12.68% to 25.07%, respectively; the roots water content of seedlings inoculated with different ecotypes of *C. geophilum* increased significantly (21.17% to 33.46%) after 7 days of drought stress treatment except for Chcg57 (Figure 3a,b). Notably, the shoot and root water content for seedlings inoculated with Jacg121 showed the highest increase of 25.07% and 33.46%, respectively.

Inoculation with most strains of *C. geophilum*, except for Jacg243 with the strongest drought-tolerant mycelia, also had a positive impact on the photosynthetic performance of the seedlings of *P. massoniana* when compared to non-inoculated seedlings after 7 days of drought stress treatment (Table 1), especially for Jacg121-inoculated and Jacg37-inoculated seedlings. In comparison to non-inoculated seedlings, the Pn, Gs, Ci, and Tr contents of Jacg121-inoculated and Jacg37-inoculated seedlings remarkably increased by 230.37%, 496.68%, 87.47%, 142.22% and 196.57%, 366.28%, 73.51%, 85.76%, respectively.

### 3.3. Effects of Inoculation with Different Ecotypic Strains of C. geophilum on Physiological Indexes of P. massoniana Seedlings during the Early Stage of Drought Stress

Compared to non-inoculated seedlings, the Pro contents in both the shoots and roots of seedlings inoculated by all strains of *C. geophilum* increased after 7 days of drought stress treatment (Figure 4a), especially the seedlings inoculated with Jacg121 with a 360.25% and 361.29% increase in Pro content in shoot and root, respectively. The shoots and roots of seedlings inoculated with Jacg121 and Jacg189 remarkably increased in SP content (shoots, 11.19% and 9.81%; roots, 16.71% and 10.79%) (Figure 4b). The SP content in the roots of seedlings inoculated with Jacg21 and Jacg16 increased 10.58% and 8.63%, respectively.

For MDA, the inoculation of Jacg21, Jacg81, Jacg189, Jacg243, and Chcg57 did not affect the contents in shoots of seedlings after 7 days of drought stress treatment, while inoculation of Jacg16, Jacg37, and Jacg121 significantly decreased the MDA content in shoots and roots of seedlings compared to the non-inoculated seedlings (Figure 4c).

The responses of antioxidant enzyme activities (CAT, POD, and SOD) in seedlings of *P. massoniana* to drought stress treatment of 7 days are shown in Figure 4d-f. In contrast to the non-inoculated seedlings, the inoculation of Jacg81 and Jacg121 resulted in a significant decrease in CAT activity of seedlings in both the shoots (61.38% and 52.46%) and roots (53.44% and 47.44%) (Figure 4d). The CAT activity in the shoots of seedlings inoculated with Jacg37 showed a remarkable decrease of 45.79%, and in the roots of seedlings inoculated with Jacg21, the CAT activity was significantly reduced by 39.33%. For POD activity, the inoculation of most strains of *C. geophilum* did not affect POD activity in shoots, while inoculation of Jacg121 and Jacg243 significantly increased POD activity (Figure 4e). In the roots, the inoculation of seedlings with Jacg121, Jacg243, Jacg37, and Jacg189 led to a significant increase in POD activity. The inoculation of most strains of *C. geophilum* also had a positive impact on SOD activity in seedlings. The SOD activity in shoots and roots was significantly increased by inoculation of Jacg121, Jacg81, and Jacg37, and inoculation with Jacg189 significantly increased SOD activity in shoots when compared to the non-inoculated seedlings (Figure 4f).

### 3.4. Comprehensive Evaluation of Drought Tolerance in P. massoniana Mycorrhizal Seedlings Inoculated by Various Ecotypes of C. geophilum

The combined responses of a total of 18 indexes including water contents (2), photosynthetic parameters (4), and physiological factors (12) in *P. massoniana* seedlings inoculated and non-inoculated by 8 ecotypic strains of *C. geophilum* after 7 days of drought stress treatment were analyzed by PCA, as reported by Zou et al. [34]. The result showed that the contribution rate of the first three comprehensive indexes (*CI_1_*, *CI_2_*, *CI_3_*) was 68.426%, 12.359%, and 7.722%, respectively, with a total of 88.507% (Appendix A). Therefore, this comprehensive evaluation of the combined responses of multiple indexes was employed to assess the drought tolerance of *P. massoniana* seedlings inoculated by each ecotypic strain of *C. geophilum*. The comprehensive evaluation of the drought tolerance level (*D* value) of the seedlings inoculated by each strain of *C. geophilum* was higher than that of non-inoculated seedlings (CK) (Table 2), indicating that the inoculation of all strains of *C. geophilum* could enhance the drought tolerance of *P. massoniana* seedlings. Of these *C. geophilum-*inoculated seedlings, the Jacg121-inoculated seedlings showed drought tolerance, while those inoculated by Chcg57 were drought-sensitive.

### 3.5. Transcriptional Responses in C. geophilum-Inoculated P. massoniana Seedlings with Various Drought Tolerance Levels under the Early Stage of Drought Stress

To evaluate the transcriptional responses of *P. massoniana* seedlings with various drought tolerance levels inoculated with *C. geophilum* after 7 days of drought stress treatment, we selected the Jacg121 (drought-tolerant)- and Chcg57 (drought-sensitive)-inoculated *P. massoniana* seedlings for RNA sequencing (Appendix A). Compared to the seedlings of non-drought stress, 4113 DEGs (upregulated: 1935; downregulated: 2178) in shoots and 7014 DEGs (upregulated: 2398; downregulated: 4616) in roots of Jacg121-inoculated seedlings were identified, respectively (Figure 5a,c and Appendix A). Similarly, the Chcg57-inoculated seedlings exhibited 3174 DEGs (upregulated: 1163; downregulated: 2011) in shoots and 3001 DEGs (upregulated: 1615; downregulated: 1386) in roots compared to non-drought stressed seedlings (Figure 5b,d and Appendix A). Of these DEGs, 1946 were shared in the shoots, while 1343 were common in the roots of Jacg121- and Chcg57-inoculated seedlings, respectively (Figure 5e,f).

To pinpoint the potential functions of the identified DEGs in Jacg121- and Chcg57-inoculated seedlings under the early stage of drought stress, the GO term enrichment analysis was performed (Figure 6 and Figure 7). In terms of biological process, the terms of DEGs enrichment in the shoots and roots of both Jacg121- and Chcg57-inoculated seedlings in response to drought were mainly associated with “metabolic process”, “single-organism process”, “cellular process”, “biological regulation”, and “response to stimulus”. As for cellular components, DEGs responsive to drought were primarily enriched in “membrane”, “cell”, and “organelle”. In terms of molecular function, the enrichment was mainly observed in “binding” and “catalytic activity”. Notably, the DEGs of shoots and roots of seedlings inoculated with Jacg121 and Chcg57 were also significantly enriched in the “antioxidant activity” of molecular function, indicating that the DEGs induced by mycorrhizal seedlings in the early stage of drought stress were mostly related to metabolism and reduction–oxidation reaction.

The KEGG pathway enrichment analysis was further performed to identify the active metabolic pathways involved in drought stress responses (Figure 8). Under the early stage of drought stress, pathways such as flavonoid biosynthesis (ko00941), plant hormone signal transduction (ko04075), photosynthesis (ko00195), mitogen-activated protein kinase (MAPK) signaling pathway-plant (ko04016), phenylpropanoid biosynthesis (ko00940), and starch and sucrose metabolism (ko00500) were significantly enriched in the shoots of both Jacg121- and Chcg57-inoculated seedlings. The response of these two kinds of mycorrhizal seedlings shoots to drought stress was mainly related to the synthesis of secondary metabolites (phenylpropanoid) and energy metabolism (photosynthesis, starch and sucrose metabolism). In the roots of drought-tolerant mycorrhizal seedlings, the top two pathways were phenylalanine metabolism (ko00360) and selenocompound metabolism (ko00450), followed by cysteine and methionine metabolism (ko00270) and isoquinoline alkaloid biosynthesis (ko00950). The DEGs of drought-sensitive mycorrhizal seedlings were mainly enriched in pathways such as pentose and glucuronate interconversions (ko00040), flavonoid biosynthesis (ko00941), starch and sucrose metabolism (ko00500), and DNA replication (ko03030). These results showed that the response in roots of drought-tolerant mycorrhizal seedlings to drought stress was related to the improvement of plant resistance and the removal of reactive oxygen species (ROS), while the synthesis of secondary metabolites and energy metabolism played a more important role in the roots of drought-sensitive mycorrhizal seedlings.

### 3.6. Analysis of Key Genes for Drought Tolerance of P. massoniana Seedlings Inoculated by C. geophilum under the Early Stage of Drought Stress

Combined with the KEGG pathway, we selected the significantly up/downregulated DEGs of the Jacg121- and Chcg57-inoculated seedlings in phenylpropanoid biosynthesis (ko00940), flavonoid biosynthesis (ko00941), glutathione metabolism (ko00480), and phenylalanine metabolism (ko00360) pathways to draw a heatmap to screen out the key genes of *P. massoniana* seedlings in response to the early stage of drought stress (Figure 9). A total of 11 common DEGs were selected in the phenylpropanoid biosynthesis pathway. Among them, 7 DEGs were significantly downregulated, and TRINITY_DN11509_c1_g1, TRINITY_DN20940_c0_g2, and TRINITY_DN1881_c1_g1 were significantly upregulated in the shoots and roots of the drought-tolerant and drought-sensitive mycorrhizal seedlings under the early stage of drought stress. Seven DEGs were selected in the flavonoid biosynthesis pathway. Except for TRINITY_DN18667_c0_g1 and TRINITY_DN1881_c1_g1, the expression levels of the remaining 5 DEGs were significantly downregulated in the shoots and roots of drought-sensitive mycorrhizal seedlings and significantly upregulated in shoots and significantly downregulated in roots of drought-tolerant mycorrhizal seedlings. Five DEGs were selected in the glutathione metabolism pathway. Except for TRINITY_DN_1340_c0_g2, the other 4 DEGs were significantly downregulated in the shoots and roots of the two kinds of mycorrhizal seedlings under the early stage of drought stress. Six DEGs were selected in the phenylalanine metabolism pathway. Under the early stage of drought stress, TRINITY_DN1073_c0_g1, TRINITY_DN1104_c0_g1, and TRINITY_DN246562_c0_g2 were significantly downregulated in the shoots and roots of the drought-tolerant and drought-sensitive mycorrhizal seedlings. The expression of TRINITY_DN14578_c0_g1 and TRINITY_DN7317_c0_g1 in shoots of the two kinds of mycorrhizal seedlings was significantly upregulated.

### 3.7. RT-qPCR Verification of DEGs

In order to verify the reliability of the sequencing results, we selected two genes in the phenylpropanoid biosynthesis pathway (TRINITY_DN3930_c0_g1, TRINITY_DN20940_c0_g2), three genes in the flavonoid biosynthesis metabolic pathway (TRINITY_DN40689_c0_g2, TRINITY_DN12846_c0_g3, TRINITY_DN18667_c0_g1), and one gene in the glutathione metabolism pathway (TRINITY_DN6096_c0_g1) for RT-qPCR validation, and the primer sequence were presented in Appendix A. As shown in Figure 10, the RNA sequencing results were consistent with the expression profiles of the TRINITY_DN20940_c0_g2, TRINITY_DN40689_c0_g2, TRINITY_DN12846_c0_g3, TRINITY_DN18667_c0_g1, and TRINITY_DN6096_c0_g1 genes.

## 4. Discussion

The current literature on enhancing the drought resistance of *P. massoniana* through the use of ECMF has predominantly concentrated on various ECMF species [35,36]. However, there has been limited attention dedicated to investigating the impact of different ecotypes within each ECMF species. *Cenococcum geophilum*, characterized by abundant genetic diversity, holds ecological significance linked to its genetic variations [21]. Numerous studies have highlighted the varying resistance levels of different ecotypic strains of *C. geophilum* to stresses such as drought, high temperature, heavy metals, and salt [23,37,38,39]. Nevertheless, existing studies primarily focus on the strain’s inherent stress resistance mechanism, with limited exploration of the symbiotic relationship between strains and plants in facing diverse stresses. In this study, we delved into the enhancement and mechanisms of drought resistance in *P. massoniana* through the inoculation of eight ecotypes of *C. geophilum*, each with distinct drought tolerances during the early stages of drought stress. Notably, in mycelial culture experiments, Jacg243 exhibited the strongest drought resistance, Chcg57 displayed the weakest, and the remaining six ecotypic strains showed no evident growth response to drought stress. Intriguingly, our findings suggest that the inoculation of all eight ecotypes effectively enhanced the drought resistance of *P. massoniana* seedlings, regardless of the drought tolerance exhibited by *C. geophilum* mycelia.

During the early stage of drought stress, the inoculation of most ecotypes of *C. geophilum* increased both shoot and root water contents, photosynthetic levels, and the contents of Pro and SP in *P. massoniana* seedlings. These responses align with previous studies emphasizing the importance of maintaining high water content, enhancing photosynthetic activity, and accumulating osmotic regulators for plant drought resistance [40]. In a broad overview, the hyphae of ectomycorrhizal (ECM) fungi exhibit rapid growth, extending several meters in the soil and covering a broader expanse compared to plant root hairs. Furthermore, ECM hyphae, being finer than root hairs, possess the capability to penetrate small gaps that are inaccessible to plant roots [41]. These traits play a crucial role in enhancing water and nutrient absorption by the host plants. Inoculation of ECMF enhances available water and stomatal conductance in host plants, ultimately enhancing photosynthetic efficiency. This improvement leads to increased carbohydrate production, thereby supporting the growth and development of the plants [42]. Additionally, as ECMF relies on the carbohydrates produced by host plants for its survival, this symbiotic relationship allows ECMF to acquire more carbohydrates from plants, ensuring the energy necessary for its growth [43]. Photosynthetic indexes not only affect the water potential of stomatal guard cells, induce water absorption or water loss, and regulate leaves’ stomatal opening and closing, but they also carry out photosynthesis through gas exchange between the internal and external environment to ensure energy supply [44]. As the main osmotic regulators of plants, Pro and SP are hydrophilic organic solvents, which can stabilize protein structure, protect macromolecular substances, and are positively correlated with the drought tolerance of plants [45]. Wang et al. [46] showed that inoculation with ECMF increased the water content, photosynthetic rate, and accumulation of osmotic adjustment substances in *P. tabulaeformis* under drought stress. Rasouli et al. [42] also reported that inoculation with mycorrhizal fungi can help *Satureja hortensis* resist drought stress by increasing its water content, photosynthetic levels, and accumulation of osmotic regulators. KEGG pathway enrichment analysis highlighted the significance of pathways such as phenylpropanoid biosynthesis, photosynthesis, and starch and sucrose metabolism in both drought-tolerant and drought-sensitive mycorrhizal seedlings. Mckiernan et al. [47] have reported that the synthesis and accumulation of phenylpropanoids can enhance the thickness of the needle cell wall and reduce the loss of water transpiration to resist the negative effects of drought stress. Feng et al. [48] have shown that genes such as *DHNs*, *LEA*, *Annexin D2*, and *NAC* in the phenylpropanoid biosynthesis pathway play an important role in protecting plant cell membrane permeability. These results further demonstrated that inoculation with *C. geophilum* could help *P. massoniana* resist drought stress by regulating energy supply and reducing water loss.

Plants have evolved complex defense mechanisms to resist external stress, which can effectively remove the accumulation of ROS and reduce the damage caused by membrane lipid peroxidation to plants [49]. Antioxidant enzymes, as an important protective enzyme in plants, play an important role in plant stress resistance, by eliminating excessive ROS, reducing cell membrane damage, and protecting cell membranes’ integrity [50]. Our results showed that the activities of POD and SOD in *P. massoniana* seedlings inoculated with *C. geophilum* were generally higher compared to non-inoculation during the early stage of drought stress, while the inoculation seedlings exhibited lower MDA content, indicating that *C. geophilum* inoculation can help *P. massoniana* to resist drought stress by reducing cell damage and scavenging ROS. Alvarez et al. [51] have reported that ECMF inoculation can influence the activity of ROS scavenging enzyme, thereby regulating host resistance to stress. Zhao et al. [19] also found that ECMF inoculation could promote SOD activity and reduce MDA content in *P. sylvestris*, which was consistent with our research results. Pan et al. [52] also suggested that plants exhibiting lower MDA content under stress had stronger drought tolerance. The results of transcriptome showed that the DEGs in *P. massoniana* seedlings inoculated with *C. geophilum* during the early stage of drought stress were significantly enriched in pathways such as flavonoid biosynthesis, plant hormone signal transduction, and MAPK signaling pathway-plant. Flavonoids are mainly derived from the phenylpropanoid pathway, which can enhance the antioxidant activity of plants and help plants resist various biotic and abiotic stresses [53]. Moreover, Hodaei et al. [54] have shown that drought stress can lead to a significant upregulation of flavonoid metabolism-related genes in *Chrysanthemum morifolium* and a significant increase in flavonoid compounds, thereby improving the antioxidant capacity of *C. morifolium*. The MAPK signaling pathway is an important tool for plants to respond to abiotic and biotic stress, with ROS accumulation serving as an activator of this pathway [55]. Liu et al. [56] found that GhMAPKKK49 in *Gossypium hirsutum* was induced by abscisic acid (ABA) and ROS and involved in ROS- and ABA-mediated responses to various abiotic stresses. Zhu et al. [57] reported that upregulation of StMAPK11 in *Solanum tuberosum* under drought conditions can promote the activity of antioxidant enzymes, thus improving drought resistance. In our study, the CAT activity of *P. massoniana* seedlings inoculated with most ecotypic strains of *C. geophilum* was significantly lower than that of non-inoculation, possibly due to the competitive relationship between CAT and POD on the substrate H_2_O_2_ [58]. However, our results showed that the POD activity in *P. massoniana* seedlings inoculated with *C. geophilum* was higher than that of non-inoculation, indicating that *C. geophilum* mainly decomposed H_2_O_2_ by increasing the POD activity of *P. massoniana*. 

In this study, we found that the drought resistance of mycorrhizal seedlings did not correlate with the inherent drought resistance of the *C. geophilum* strain itself. Jacg121, despite showing no growth response to drought in mycelium culture, exhibited the strongest drought resistance when forming symbionts with *P. massoniana*. Conversely, the mycorrhizal seedlings of *P. massoniana* inoculated with Jacg243 of the strongest drought resistance in mycelium culture demonstrated comparatively weak drought resistance. Under the same water condition, the biomass of *P. massoniana* seedlings inoculated with Jacg121 was significantly higher than that of Jacg243, especially the fresh and dry weights of roots. This implies that the establishment of a symbiotic relationship with Jacg121 significantly increased the root absorption area, enhancing the seedlings’ moisture absorption capacity. Rasouli et al. [42] have already reported that ECMF can expand the root absorption area by establishing a symbiotic relationship, improve the water status and nutrient metabolism of host plants, and avoid or slow down the drought stress to plants. This emphasizes the importance of promoting root growth to help plants overcome drought stress [59,60]. In our research, we isolated strains of *C. geophilum* from the tropical region of Thailand [61]. Subsequently, we investigated the impact of temperature on the mycelial growth of *C. geophilum* [37]. Our findings indicate that certain strains of *C. geophilum* exhibit resistance to high temperatures. Furthermore, our research group explored the responses of *C. geophilum* to salt and cadmium exposure [38,39,62]. The results revealed that *C. geophilum* demonstrates resilience to both salt and cadmium stressors. Consequently, these findings suggest that *C. geophilum* holds significant potential for applications in reforestation efforts under severe stress conditions including severe droughts induced by climate change. Based on our findings, it is suggested to cultivate drought-resistant ECM seedlings using strains that not only exhibit drought resistance but also possess the ability to significantly enhance plant growth in mycorrhizal afforestation in drought areas.

## 5. Conclusions

This study revealed that the inoculation of various ecotypes of *C. geophilum* had a positive impact on both shoot and root water contents, photosynthesis, osmotic adjustment substance accumulations, and antioxidant enzyme activities in *P. massoniana* seedlings, offering varying degrees of support against early-stage drought stress. However, the level of drought resistance of mycorrhizal seedlings of *P. massoniana* did not correlate with the inherent drought resistance of the *C. geophilum* strain itself. The strains of *C. geophilum* that exhibit greater efficacy in enhancing the growth, particularly the development of roots in *P. massoniana* seedlings, demonstrate superior capabilities in augmenting the drought resistance of these seedlings during early-stage drought stress, indicating that, by applying ECMF to reforest vegetation in drought areas, the species that exhibit both drought resistance and the ability to significantly enhance plant growth are available.

## Figures and Tables

**Figure 1 jof-10-00071-f001:**
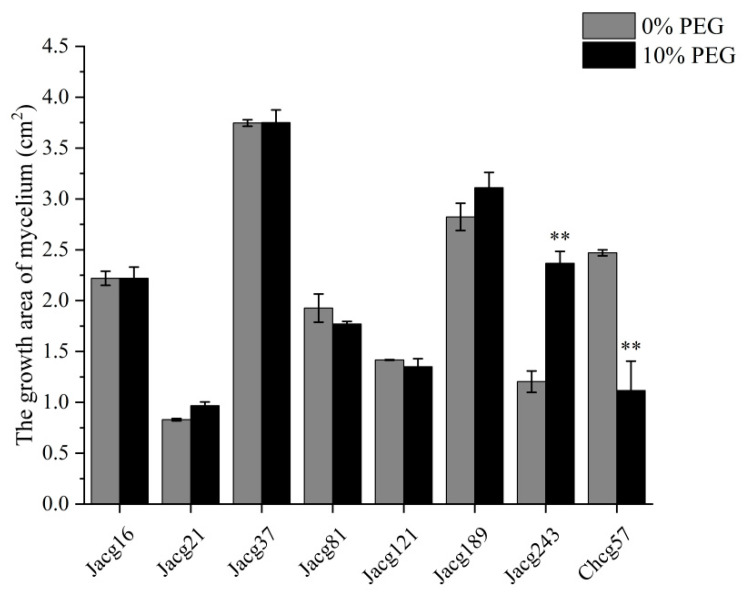
The mycelial growth of eight ecotypes of *Cenococcum geophilum* under drought and non-drought treatments. Data and bars are shown as mean and ± SE of the replicates, respectively (n = 3). The statistically significant difference between 0% PEG and 10% PEG treatments is tested by Student’s *t*-test (** *p* < 0.01).

**Figure 2 jof-10-00071-f002:**
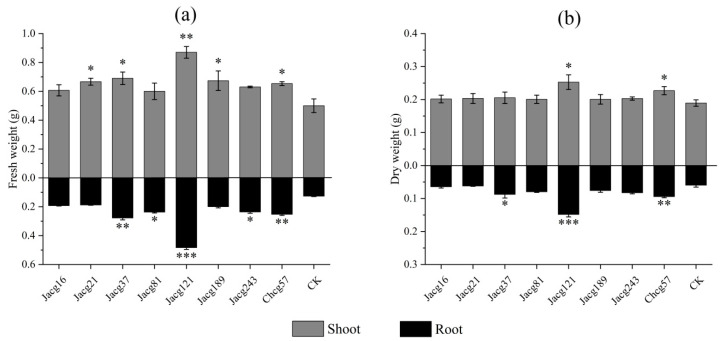
Effects of different *Cenococcum geophilum* ecotypes inoculation on (**a**) fresh weights and (**b**) dry weights of *Pinus massoniana* seedlings. Data and bars are shown as the mean and ± SE of the replicates, respectively (n = 3). The statistically significant difference between different *Cenococcum geophilum* ecotypes inoculation and non-inoculation treatments was tested by the Mann-Whitney U test (* *p* < 0.05; ** *p* < 0.01; *** *p <* 0.001). CK, non-inoculation.

**Figure 3 jof-10-00071-f003:**
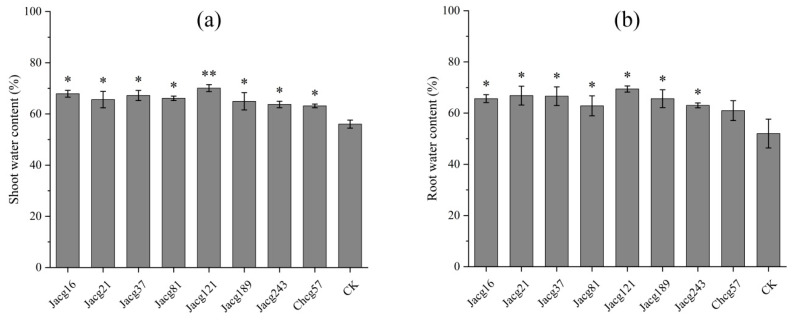
Effects of different *Cenococcum geophilum* ecotypes inoculation on the (**a**) shoot and (**b**) root water contents of *Pinus massoniana* seedlings after 7 days of drought stress treatment (field capacity 30-35%). Data and bars are shown as the mean and ± SE of the replicates, respectively (n = 3). The statistically significant difference between different *C. geophilum* ecotypes inoculation and non-inoculation treatments was tested by the Mann-Whitney U test (* *p* < 0.05; ** *p* < 0.01). CK, non-inoculated.

**Figure 4 jof-10-00071-f004:**
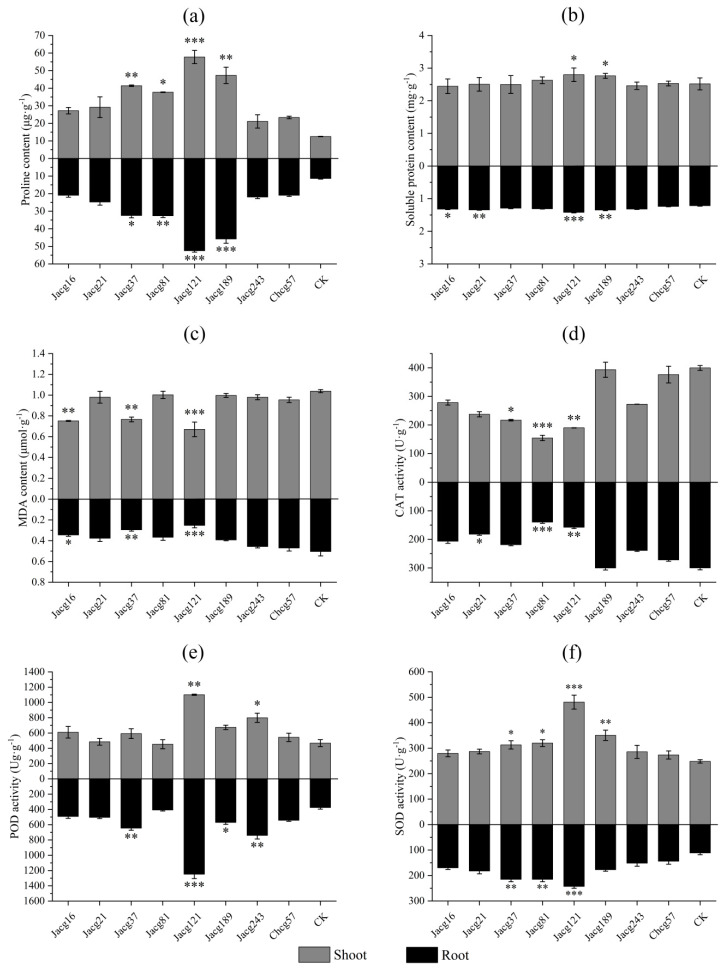
Effects of different *Cenococcum geophilum* ecotypes inoculation on the (**a**) proline content, (**b**) soluble protein content, (**c**) malondialdehyde (MDA) content, (**d**) catalase (CAT) activities, (**e**) peroxidase (POD) activities, and (**f**) superoxide dismutase (SOD) activities of *Pinus massoniana* seedlings after 7 days of drought stress treatment (field capacity 30-35%). Data and bars are shown as the mean and ± SE of the replicates, respectively (n = 3). The statistically significant difference between different *C. geophilum* ecotypes inoculation and non-inoculation treatments was tested by the Mann-Whitney U test (* *p* < 0.05; ** *p* < 0.01; *** *p* < 0.001). CK, non-inoculation.

**Figure 5 jof-10-00071-f005:**
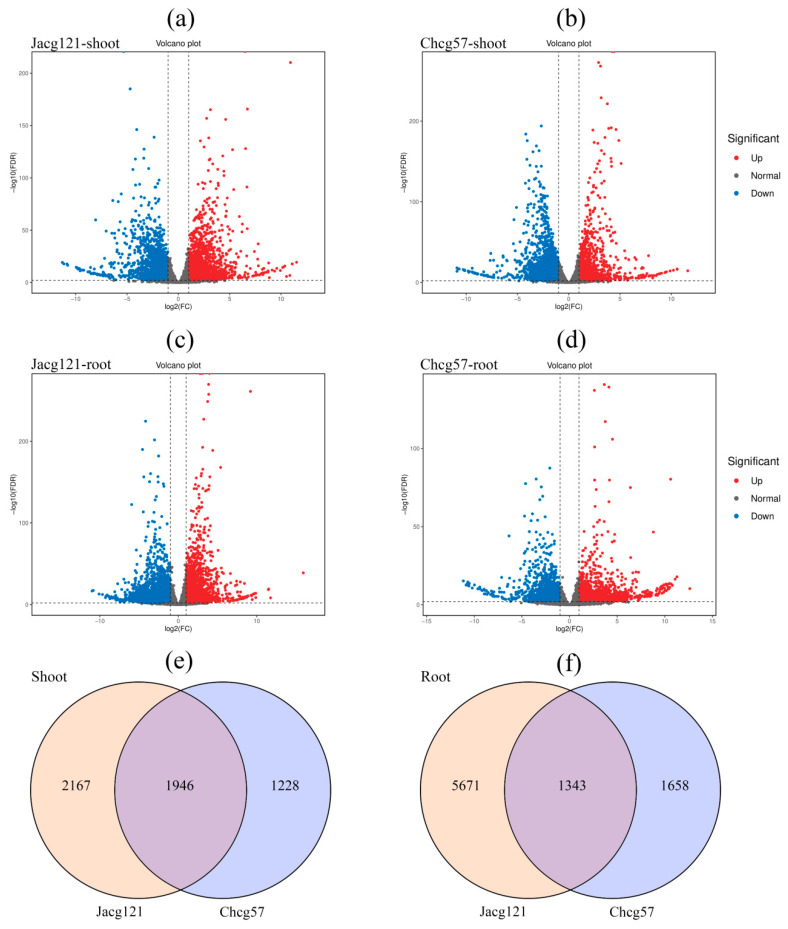
Volcano (**a**–**d**) and Wenn (**e**,**f**) maps of differentially expressed genes (DEGs) analysis of drought-tolerant (Jacg121) and drought-sensitive (Chcg57) seedlings of *Pinus massoniana* inoculated by *Cenococcum geophilum* after 7 days of drought (field capacity 30–35%) and well-watered (field capacity 85–90%) treatments. The red spot indicates significantly upregulated genes after drought treatment; the blue spot indicates significantly downregulated genes; and the gray spot indicates no-change genes. |log2Foldchange| ≥ 2 are used as the screening criteria of DEGs.

**Figure 6 jof-10-00071-f006:**
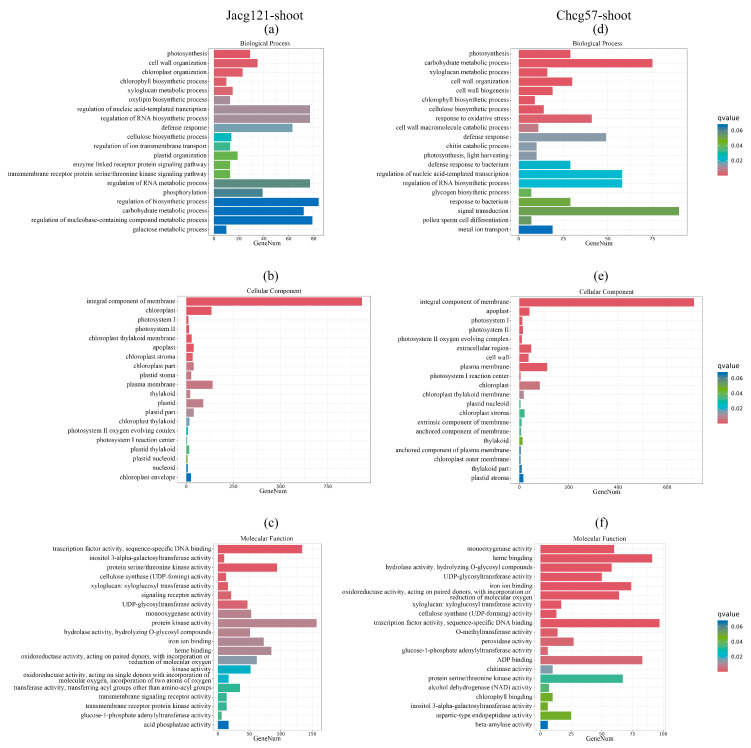
Gene ontology (GO) enrichment analysis of the differentially expressed genes (DEGs) in shoots of drought-tolerant (Jacg121) (**a**–**c**) and drought-sensitive (Chcg57) (**d**–**f**) seedlings of *Pinus massoniana* inoculated by *Cenococcum geophilum* after 7 days of drought (field capacity 30–35%) and well-watered (field capacity 85–90%) treatments. The abscissa is the number of genes, and the ordinate is the enriched GO terms.

**Figure 7 jof-10-00071-f007:**
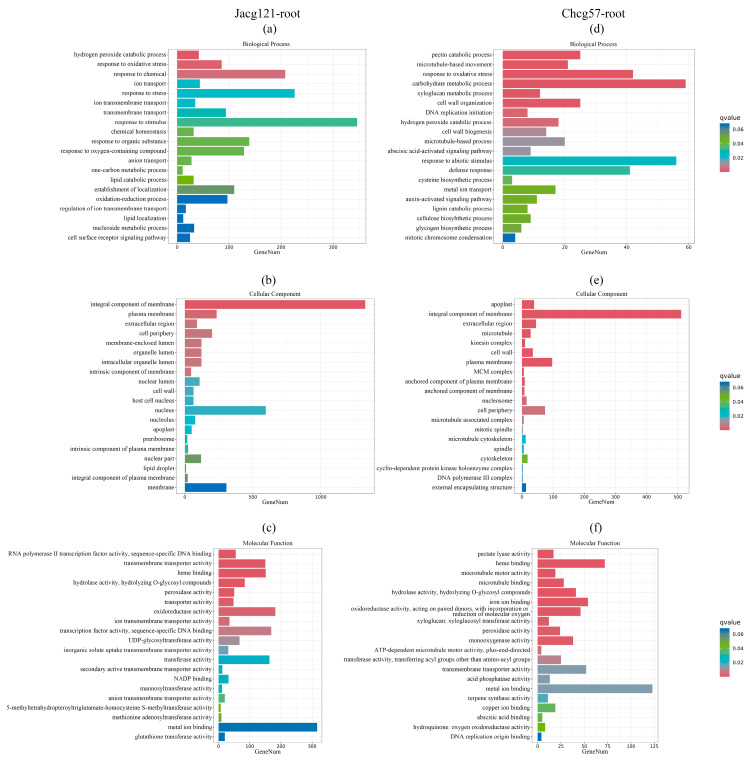
Gene ontology (GO) enrichment analysis of the differentially expressed genes (DEGs) in roots of drought-tolerant (Jacg121) (**a**–**c**) and drought-sensitive (Chcg57) (**d**–**f**) seedlings of *Pinus massoniana* inoculated by *Cenococcum geophilum* after 7 days of drought (field capacity 30–35%) and well-watered (field capacity 85–90%) treatments. The abscissa is the number of genes, and the ordinate is the enriched GO terms.

**Figure 8 jof-10-00071-f008:**
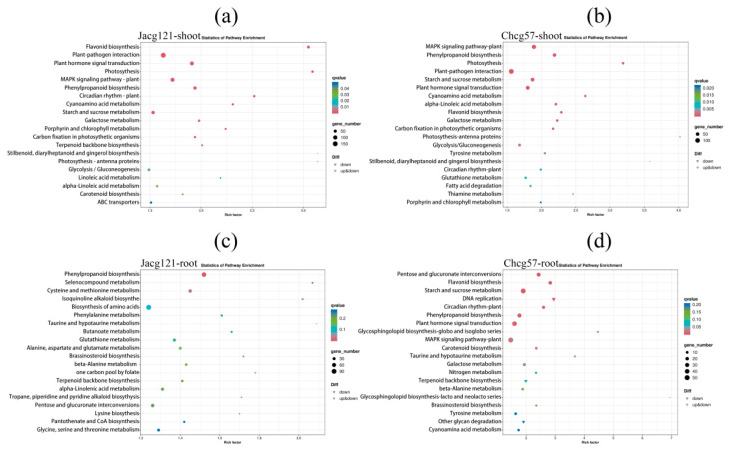
The Kyoto Encyclopedia of Genes and Genomes (KEGG) enrichment analysis of differentially expressed genes (DEGs) in shoots and roots of drought-tolerant (Jacg121) (**a**,**c**) and drought-sensitive (Chcg57) (**b**,**d**) seedlings of *Pinus massoniana* inoculated by *Cenococcum geophilum* after 7 days of drought (field capacity 30–35%) and well-watered (field capacity 85–90%) treatments. The size of the symbol represents the number of DEGs involved in the corresponding pathway.

**Figure 9 jof-10-00071-f009:**
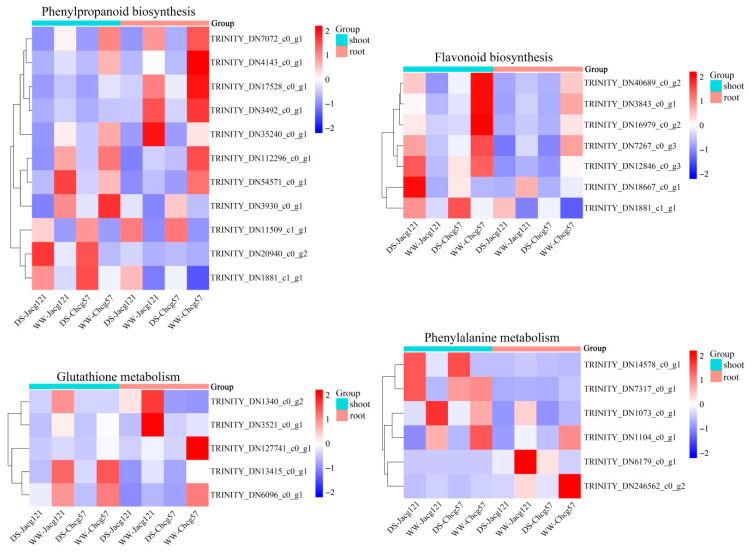
The heatmaps of partly upregulated or downregulated differentially expressed genes (DEGs) in different pathways of drought-tolerant (Jacg121) and drought-sensitive (Chcg57) seedlings of *Pinus massoniana* inoculated by *Cenococcum geophilum* after 7 days of drought (field capacity 30–35%) and well-watered (field capacity 85–90%) treatments. The Y- and X-axes represent the DEGs and different samples, respectively. The different colors of the heatmaps, ranging from blue over white to red, represent scaled expression levels of genes with [log2 (FPKM + 1)] across different samples. DS, drought stress; WW, well-watered.

**Figure 10 jof-10-00071-f010:**
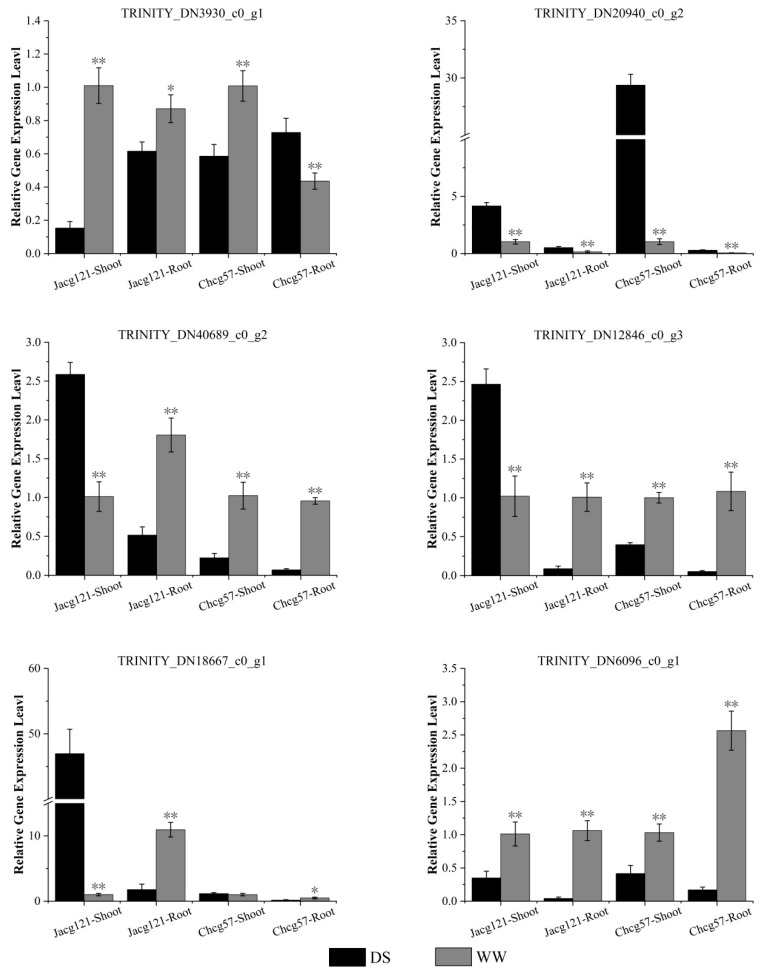
The relative expression levels of six candidate genes in drought-tolerant (Jacg121) and drought-sensitive (Chcg57) seedlings of *Pinus massoniana* inoculated by *Cenococcum geophilum* were analyzed by RT-qPCR after 7 days of drought (field capacity 30–35%) and well-watered (field capacity 85–90%) treatments. The relative expression changes of candidate genes in different treatments were calculated by the 2^−△△Ct^ method and expressed as mean and ± SE (n = 3). The statistically significant difference between drought and well-watered treatments was tested by the Student’s *t*-test (* *p* < 0.05; ** *p* < 0.01). WW, well-watered; DS, drought stress.

**Table 1 jof-10-00071-t001:** Effects of inoculation with different *Cenococcum geophilum* ecotypes on photosynthetic parameters of *Pinus massoniana* seedlings after 7 days of drought stress treatment (field capacity 30–35%).

ID	Pn (μmol·m^−2^·s^−1^)	Gs (mol·m^−2^·s^−1^)	Ci (μmol·m^−2^·s^−1^)	Tr (mmol·m^−2^·s^−1^)
Jacg16	1.57 ± 0.06	0.023 ± 0.001	322.17 ± 4.37 ***	0.95 ± 0.01
Jacg21	2.45 ± 0.32 *	0.019 ± 0.004	214.01 ± 3.12	1.00 ± 0.02
Jacg37	3.22 ± 0.03 **	0.036 ± 0.001 ***	280.97 ± 1.73 *	1.72 ± 0.01 ***
Jacg81	3.53 ± 0.08 ***	0.026 ± 0.007 *	207.72 ± 8.04	1.16 ± 0.03 *
Jacg121	3.58 ± 0.12 ***	0.046 ± 0.003 ***	303.57 ± 4.77 **	2.25 ± 0.01 ***
Jacg189	1.93 ± 0.02	0.035 ± 0.003 **	327.84 ± 0.94 ***	1.59 ± 0.01 **
Jacg243	1.47 ± 0.04	0.020 ± 0.001	261.70 ± 3.00	0.98 ± 0.01
Chcg57	2.30 ± 0.08	0.024 ± 0.007	280.65 ± 1.61 *	1.20 ± 0.03 *
CK	1.08 ± 0.01	0.008 ± 0.001	161.93 ± 3.55	0.93 ± 0.01

Note: Data are expressed as mean ± SE (n = 3). The statistically significant difference between different *C. geophilum* ecotypes inoculation and non-inoculation treatments was tested by the Mann-Whitney U test (* *p* < 0.05; ** *p* < 0.01; *** *p* < 0.001). Pn, net photosynthetic rate; Gs, stomatal conductance; Ci, intercellular CO_2_ concentration; Tr, transpiration rate; CK, non-inoculation.

**Table 2 jof-10-00071-t002:** The value of comprehensive index (*CI*), index weight (*W_j_*), *µ*(*X_j_*), *D* value (comprehensive evaluation of drought tolerance), and comprehensive evaluation in *Pinus massoniana* seedlings inoculated by different *Cenococcum geophilum* ecotypes after 7 days of drought stress (field capacity 30–35%).

Code	*CI_1_*	*CI_2_*	*CI_3_*	*μ*(*X_1_*)	*μ*(*X_2_*)	*μ*(*X_3_*)	*D*	Comprehensive Evaluation *
Jacg16	−0.083	−0.382	−1.990	0.437	0.354	0.000	0.386	6
Jacg21	−0.118	−1.057	−0.437	0.427	0.154	0.450	0.391	5
Jacg37	0.555	−0.492	−0.107	0.610	0.321	0.545	0.564	3
Jacg81	0.188	−1.574	1.465	0.510	0.000	1.000	0.483	4
Jacg121	1.991	0.511	0.417	1.000	0.619	0.697	0.919	1
Jacg189	0.319	1.792	0.337	0.546	1.000	0.673	0.621	2
Jacg243	−0.454	0.230	−0.627	0.336	0.536	0.394	0.369	7
Chcg57	−0.704	0.580	−0.060	0.268	0.640	0.559	0.347	8
CK	−1.693	0.393	1.002	0.000	0.584	0.866	0.160	9
Index weight (*w_j_*)				0.770	0.140	0.09		

Note: CK, non-inoculation; *, the lower level indicates the higher drought tolerance.

## Data Availability

The data presented in this study are available within the article and Appendix A. Data for sequence reads are available in a publicly available repository (NCBI), reference number PRJNA1048730.

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
