# Peer review of "Physiological and Transcriptome Responses of Pinus massoniana Seedlings Inoculated by Various Ecotypes of the Ectomycorrhizal Fungus Cenococcum geophilum during the Early Stage of Drought Stress"

_jof, 2024, doi:10.3390/jof10010071_

Round 1
Reviewer 1 Report
Comments and Suggestions for Authors
Dear authors,
Please add your answers to the following questions to the discussion:
Is the effectiveness of the fungi used for mycorrhizal treatment due to their rapid growth in the soil and thus to the multiplication of the reach of the root systems of the pines?
Did this allow them to extract more water with mineral salts?
Did the increase in available water increase the efficiency of photosynthesis? Did more assimilates favour plant growth, root nutrition and symbiosis of the resulting fungal associations?
How might climate change, particularly the rise in temperature, affect the mycorrhizae produced? In other words, could mycorrhisation of C. geophilum (a widespread fungus that readily forms associations with numerous plant species) be a future tool in forestry and agriculture against the predicted severe droughts?
If mycorrhisation can affect the phenolic association, synthesis and accumulation of phenylpropanoids, can it also increase the thickness of the cell wall of needles and reduce the loss of water transpiration to resist the negative effects of drought stress?
Comments on the Quality of English LanguageThe English is fine, a minor revision of the language is required.
Reviewer 2 Report
Comments and Suggestions for Authors
Zhang and colleagues conducted a study to assess the impact of conferring drought resistance to Punys massoniana plants by inoculating them with different ecotypes of the ectomycorrhizal fungus Cenococcum geophilum species. The study demonstrated notable improvements in various parameters such as water content, photosynthesis, and antioxidant enzyme activities in plants subjected to drought stress and inoculated with C. geophilum strains compared to the non-inoculated control. Additionally, they performed an RNAseq analysis to identify key metabolic processes and essential genes crucial for drought stress resistance. Overall, the study was well-executed, featuring meticulously planned experiments that yielded clear results.
Suggestions for Improvement:
Abstract Focus: Consider focusing the abstract on providing a concise overview of the general experiments conducted and highlighting the most relevant results, especially emphasizing the most intriguing findings obtained from the RNA-seq analysis. Remove unnecessary details, such as specific experimental design descriptions, to streamline the abstract's content.
Molecular Characterization: It's essential to clarify whether these ecotypes of C. geophilum have previously undergone molecular characterization to confirm their belonging to the same species. If this information exists and was not cited, it would be beneficial to include it. If not, conducting a molecular identification using ITS (Internal Transcribed Spacer) sequences would strengthen the study's credibility and ensure consistency when working with ecotypes.
Figure Interpretation: Figures 6 and 7 present the number of differential genes within each functional category, but it's unclear if an enrichment analysis was conducted. To enhance the figures' clarity and significance, consider performing statistical tests, such as a hypergeometric test, to identify and discuss enriched categories, elucidating which functions are vital under the evaluated conditions.
In general, the figures and the written content are clear. Addressing the suggested improvements would enhance the study's potential for publication.
